# Improving Kidney Disease Care: One Giant Leap for Nephrology

**DOI:** 10.3390/biomedicines12040828

**Published:** 2024-04-09

**Authors:** Michele Provenzano, Lilio Hu, Edoardo Tringali, Massimo Senatore, Roberta Talarico, Michele Di Dio, Chiara Ruotolo, Gaetano La Manna, Carlo Garofalo, Gianluigi Zaza

**Affiliations:** 1Department of Pharmacy Health and Nutritional Sciences, University of Calabria, 87036 Rende, Italy; m.senatore@aocs.it (M.S.); r.talarico@aocs.it (R.T.); 2Nephrology, Dialysis and Kidney Transplant Unit, IRCCS Azienda Ospedaliero-Universitaria di Bologna, 40138 Bologna, Italy; lilio.hu@studio.unibo.it (L.H.); edoardo.tringali@studio.unibo.it (E.T.); gaetano.lamanna@unibo.it (G.L.M.); 3Department of Medical and Surgical Sciences (DIMEC), Alma Mater Studiorum University of Bologna, 40138 Bologna, Italy; 4Division of Urology, Department of Surgery, SS Annunziata Hospital, 87100 Cosenza, Italy; m.didio@aocs.it; 5Unit of Nephrology, Department of Advanced Medical and Surgical Sciences, University of Campania “Luigi Vanvitelli”, 81100 Caserta, Italy; chiara.ruotolo@yahoo.it (C.R.); carlo.garofalo@unicampania.it (C.G.)

**Keywords:** CKD, albuminuria, transplant immunology, prognosis, ESKD, target therapy

## Abstract

Nephrology is an ever-evolving field of medicine. The importance of such a discipline is related to the high clinical impact of kidney disease. In fact, abnormalities of kidney function and/or structure are common in the general population, reaching an overall prevalence of about 10%. More importantly, the onset of kidney damage is related to a strikingly high risk of cardiovascular events, mortality, and progression to kidney failure which, in turn, compromises quality and duration of life. Attempts to comprehend the pathogenesis and molecular mechanisms involved in kidney disease occurrence have prompted the development and implementation of novel drugs in clinical practice with the aim of treating the ‘specific cause’ of kidney disease (including chronic kidney disease, glomerular disease, and genetic kidney disorders) and the main immunological complications following kidney transplantation. Herein, we provide an overview of the principal emerging drug classes with proved efficacy in the context of the aforementioned clinical conditions. This can represent a simplified guide for clinical nephrologists to remind them of the vast and heterogeneous armamentarium of drugs that should be used in the present and the future to improve the management of patients suffering from kidney disease.

## 1. Introduction

Chronic kidney disease (CKD) is definitely gaining momentum as a public health issue and non-communicable disease of the new millennium [1]. The onset of CKD results in, for each affected individual, a strikingly high risk of cardiovascular (CV) events, hospitalizations, all-cause mortality, and other major outcomes, including kidney failure (KF) [2]. In the last two decades, a doubling of the prevalence and incidence of CKD has been observed due to the increase in average lifespan and the raising prevalence of risk factors such as diabetes and hypertension. The chronicity of kidney disease is a significant burden for each patient and for healthcare systems. The chronic condition is generally linked to the presence of irreversible lesions in the kidney (including chronic inflammation and fibrosis) and manifests with typical clinical signs like anemia, hyperkalemia, mineral bone disorders, metabolic acidosis, and hypertension [3]. Despite this well-defined group of complications, several heterogeneous causes may determine the initiation of kidney damage. Moreover, the complexity is enhanced by the fact that most of these complications can be the cause or the consequence of kidney function impairment. The recent expansion of randomized studies enrolling thousands of patients with kidney disease gave us two very important lessons. (1) Treating complications is at least as important as treating the cause of disease, but (2) treating the specific cause of disease remains the main goal for clinical nephrologists. This concept can be extended to the field of kidney transplantation (KT), where targeted therapy was recently introduced to improve the management of acute rejection and desensitization approaches. Furthermore, the optimization of therapeutic approaches in kidney disease requires particular consideration in relation to drug pharmacokinetics and potential adverse drug reactions in this population, which could also explain the difficulties encountered in the development of new treatments [4,5].

One other important point in the field of kidney disease is how to target and monitor treatment effects. Albuminuria reduction in the first weeks after the start of treatment and the slowing of eGFR decline in the first years are considered the most important endpoints of novel drugs used in CKD patients. Other important therapeutic targets in these patients are represented by increases in hemoglobin and decreases in parathyroid hormone and serum potassium levels. The regulation of the immune system, complement activation, and the blockade of endothelin receptors are the core targets of drugs implemented in glomerular disease. Regulating the immune system and the cleavage of immunoglobulin are innovative therapies used to improve the management of KT patients. The inhibition of gene expression via RNA interference is a novel and interesting approach in the treatment of genetic kidney disease.

The aim of the present review is to summarize recent discoveries in terms of drugs that target a specific kidney disease or a specific pattern of damage, with the aim of individual protection against CV and renal risk, and to present how these new findings are changing nephrology toward a systematic and more organized discipline. The explored topics concern chronic kidney disease, glomerular disease, kidney transplantation, and genetic kidney disease.

## 2. Methods

We designed a narrative review of the literature. Article searching was conducted using PubMed, running one research study for each area of interest: chronic kidney disease, glomerular disease, kidney transplantation, and genetic kidney disease. For each area we restricted the research to ‘clinical trials’ between 1 January 2000 and 21 March 2024. We decided to report the results by area of research and by level of evidence, separately depicting high-evidence studies (mainly phase 3 clinical trials) and low–moderate-evidence studies (mainly phase 2 or weaker clinical studies). The main aim of the study was to summarize the emerging therapies in the different abovementioned areas of nephrology.

## 3. Chronic Kidney Disease

In the past few years, there has been extraordinary development and implementation in the clinical practice of novel drugs aimed at slowing the progression of kidney damage and reducing cardiovascular and all-cause mortality risk [6]. Before this, the standard-of-care treatment for CKD consisted of the non-specific inhibition of the renin–angiotensin–aldosterone system (RAAS), which was demonstrated to achieve nephroprotection associated with a reduction in blood pressure and proteinuria in the first weeks of treatment [7,8,9,10]. However, some patients with CKD do not respond to these drugs or show a partial response, thus receiving no benefit but possibly adverse effects. The advancement of knowledge has also highlighted the importance of treating the main complications of CKD, i.e., the alterations linked to the decline in kidney function such as anemia, inflammation, mineral bone disease, and in the most advanced stage, pruritus and fatigue.

### 3.1. Albuminuria-Lowering Therapies (High-Grade Evidence)

Sodium-glucose cotransporter 2 (SGLT2) inhibitors are relatively new drugs that inhibit glucose absorption in the proximal tubule of the kidney. They exert nephroprotection by reducing blood glucose levels (inducing glycosuria), downregulating glomerular hyperfiltration, and improving oxygenation of the renal parenchyma [11]. Large phase 3 clinical trials demonstrated that the SGLT2 inhibitors dapagliflozin, empagliflozin, and canagliflozin confer protection against fatal and non-fatal CV events and CKD progression over time, when administered in addition to RAAS inhibition, in patients with reduced eGFR (ranging from 20 to 90 mL/min/1.73 m^2^) and albuminuria (>200–300 mg/g) [12,13,14]. More importantly, this finding was confirmed in both diabetic and non-diabetic patients with CKD. The recently published 2024 KDIGO Clinical Practice Guidelines for the Management of CKD strongly recommend treating CKD patients (eGFR > 20 mL/min) with type 2 diabetes (T2D) or hearth failure (HF) with an SGLT2 inhibitor regardless of the presence of albuminuria; the same level of recommendation is indicated for non-diabetic CKD (eGFR > 20 mL/min) with increased albuminuria (>200 mg/g) [15].

Another class of drugs that have shown similar results are non-steroidal mineralocorticoid receptor antagonists (MRA). MRAs act through a selective and potent blockade of the MR receptor and induce a conformational change of the receptor–ligand complex, leading to the downregulation of inflammatory genes [16]. These aspects are particularly innovative and combine the effects of MR antagonism, the anti-inflammatory properties, and the low incidence of particularly fearful side effects related to the classic steroidal MRA (i.e., gynecomastia and hyperkalemia). Two major phase 3 studies have shown that the MRA finerenone reduces CV risk (especially hospitalization for heart failure) and risk for CKD progression by about 20% in patients with albuminuric CKD and type 2 diabetes [17,18], leading to grade 1A recommendation by the 2023 Update of the ESC Guidelines for Hearth Failure [19]. Similarly, the 2024 KDIGO Guidelines on CKD management [15] suggest the use on non-steroidal MRA in patients with T2D and CKD (eGFR > 20 mL/min) and persistent albuminuria (>30 mg/g) despite maximal RAAS inhibitor treatment.

In type 2 diabetes, the combination of drugs that simultaneously reduce residual CV risk and CKD progression and enable a better control of diabetes per se has been eagerly investigated. To this aim, a promising treatment could be represented by the receptor agonisms (RAs) for two substrates, the glucose-dependent insulinotropic polypeptide (GIP) and Glucagon-like peptide 1 (GLP-1). They stimulate glucose-dependent insulin release from the pancreatic β-cells. Tirzepatide, a dual GLP-1/GIP-RA, was more effective at slowing eGFR reduction and reducing albuminuria than antidiabetic agents such as insulin [20]. The GLP1-RA semaglutide is now under investigation in patients with CKD and type 2 diabetes given its important effect on hyperfiltration and glomerular damage, as documented in pilot studies [21,22] and confirmed by the early stop for efficacy of the FLOW-trial [23].

### 3.2. Hypoxia-Inducible Factor Stabilizers (High-Grade Evidence)

Another important step forward in the management of CKD is represented by the transition from the subcutaneous to the oral route for the treatment of anemia, a frequent complication in advanced stages of CKD [24]. Drugs that inhibit prolyl hydroxylase (PH) enzymes and, in turn, stabilize hypoxia-inducible factor (HIF) have proven similar efficacy to injectable preparations of recombinant erythropoiesis-stimulating agents (ESAs) in correcting anemia and in maintaining hemoglobin concentration levels over time [25]. Even more importantly, HIF-PHs have comparable efficacy to ESAs, not only in immediate control of anemia, but also in reducing CV risk over time, despite their more comfortable and less invasive use [26].

The emergence of these novel anti-anemic therapeutics has been addressed in a recent 2023 KDIGO Controversies Conference [27] and practical recommendations will be released in the upcoming 2024 KDIGO Guidelines on the management of CKD-related anemia.

### 3.3. Anti-Inflammatory Agents

The increase in the inflammatory milieu has always been considered an active and significant prognostic factor of kidney failure and CV risk in CKD patients. Some prognostic studies have also demonstrated these suggestions by associating the inflammatory state with a worse prognosis [28]. The following step, namely to prove that reducing inflammation may contribute to reducing the risk of events over time in CKD patients, is ambitious.

Ziltivekimab (low-grade evidence), a human monoclonal antibody targeting interleukin-6 (IL-6) ligand, was tested in CKD patients with inflammatory status and was shown to improve the markers of anemia [29], whereas Canakinumab (high-grade evidence), an antibody targeting IL-1β, has also been shown to reduce major CV events in CKD patients and atherosclerosis and, particularly, in those with immediate anti-inflammatory response to this drug [30].

### 3.4. Treatments of CKD Mineral Bone Disorders (High-Grade Evidence)

Although bisphosphonates are commonly used for osteoporosis, they are not recommended for patients with severely reduced kidney function due to the high risk of adverse effects [31]. Romosozumab, a humanized monoclonal antibody against sclerostin, an endogenous cytokine that inhibits bone formation and stimulates bone resorption, demonstrated superiority in reducing fractures compared to placebo and oral alendronate in large phase 3 clinical trials. In the Fracture Study in Postmenopausal Women with Osteoporosis (FRAME), treatment with romosozumab led to significant gains in bone mineral density (BMD) and a lower relative risk of fractures compared to placebo [32]. Similar positive outcomes were observed in the Active-Controlled Fracture Study in Postmenopausal Women with Osteoporosis at High Risk (ARCH) comparing romosozumab to alendronate [33]. The merged analysis of two trials involving a total of 11,224 patients confirmed the efficacy of romosozumab in increasing BMD and reducing the relative risk of fractures across different levels of kidney function. Adverse events and changes in kidney function were similar across baseline kidney function groups, while concern about potential MACE led to a black-box warning for patients with recent stroke or MI [34].

In the treatment of secondary hyperparathyroidism in patients with CKD, the goal is to control parathyroid hormone levels in the early stages of disease, when parathyroid cells are still sensitive to physiological 1,25(OH)2D signaling. Recent evidence suggests that maintaining 25(OH)D levels above 50 ng/mL effectively reduces parathyroid hormone levels. Guidelines recommend nutritional vitamin D supplements for CKD patients not on dialysis, reserving active vitamin D for advanced CKD or uncontrolled parathyroid hormone levels [35]. However, these interventions have limitations, including modest effects and risks of hypercalcemia. Extended-release calcifediol emerges as a promising option for intermediate stages of CKD, effectively reducing parathyroid hormone levels comparable to active vitamin D analogues. Clinical trials show dose-dependent increases in 25(OH)D and physiological increases in 1,25(OH)2D, with sustained reduction in parathyroid hormone and minimal impact on mineral balance [36]. Meanwhile, emerging data from real-world clinical experience with extended-release calcifediol for patients with stage 3 or 4 CKD suggest that it is comparable to active vitamin D analogues for controlling parathyroid hormone levels, with the added benefit of replenishing 25(OH)D [37]. In practice, patients with serum 25(OH)D levels ≥ 30 ng/mL and normal calcium, phosphorus, and parathyroid hormone may not require therapy. For those with 25(OH)D < 30 ng/mL and normal calcium, phosphorus, and parathyroid hormone, nutritional vitamin D like cholecalciferol is suggested. In cases of 25(OH)D < 30 ng/mL with elevated parathyroid hormone, extended-release calcifediol is recommended, and if persistent elevation occurs, a combination of nutritional and active vitamin D may be considered.

### 3.5. Pruritus (High-Grade Evidence)

Pruritus is associated with CKD, particularly in patients with KF or those undergoing hemodialysis (HD) [38,39]. CKD-associated pruritus (CKD-aP) is a common and often overlooked condition, affecting 26–48% of HD patients, negatively impacting their quality of life (QoL) and leading to various complications such as sleep disturbances, depression, and increased mortality [40]. Despite the importance of managing pruritus in CKD patients, treatment options are limited [41,42]. Off-label treatments such as antihistamines and corticosteroids are used, but their effectiveness can be hampered by side effects [43]. Only one treatment, nalfurafine, has been approved in Japan and South Korea. Several experimental therapies are being evaluated, but well-designed clinical studies are required to assess their efficacy and safety [44,45]. In August 2021, difelikefalin, a novel κ-opioid receptor agonist, received approval from the US Food and Drug Administration (FDA) for treating moderate to severe CKD-aP in adults undergoing HD [46]. In 2022, it also gained approval from the European Medicines Agency (EMA). The approval was based on evidence from the KALM-1 and KALM-2 phase 3 studies, in which difelikefalin demonstrated significant reductions in itch intensity compared to a placebo [47].

### 3.6. New Potassium Binders (High-Grade Evidence)

Standard-of-care treatment with ACE inhibitors and/or angiotensin receptor blockers is known to be characterized by a higher risk of hyperkalemia, which was the main reason for early discontinuation of this nephroprotective therapy. Moreover, classical potassium-lowering therapies, such as sodium polystyrene sulfonate, were not indicated for long-term use due to potential serious gastrointestinal adverse effects, such as intestinal necrosis and colitis. In recent years, patiromer and sodium zirconium cyclosilicate have been developed and recommended for the long-term prevention of recurrent hyperkalemia [48]. They are non-absorbed cation exchange polymers which increase fecal excretion of potassium and reduce gastrointestinal absorption of free potassium, leading to decreased serum potassium levels. The use of these new potassium binders has been shown to enable the optimal administration of RAAS inhibitor drugs [49].

## 4. Glomerular Disease

Glomerular diseases (also known as ‘glomerulonephritis’) are inherited or acquired disorders which manifest with urinary abnormalities, systemic signs, and symptoms leading to CKD of variable severity. The pathogenesis of glomerular disease is heterogeneous and, in most cases, may involve immunological pathways. In the past few years, a number of novel drugs have been introduced in order to treat the etiological cause of glomerular damage.

### 4.1. Belimumab (High-Grade Evidence)

One of the most common and severe manifestations of systemic lupus erythematosus (SLE) is lupus nephritis (LN), a condition that progresses to ESKD in approximately 5–20% of patients [50]. Conventional immunosuppressive treatment of LN includes glucocorticoids, azathioprine, mycophenolate mofetil, or cyclophosphamide; however, the efficacy is not uniform among patients and drug toxicity is also a major concern. To address these unmet needs, belimumab was approved by FDA in 2020 for the treatment of both SLE and active LN in adults [51]. Belimumab is an IgG-1λ monoclonal antibody that specifically targets B-lymphocyte activating factor (BAFF), also called B-lymphocyte stimulator (BlyS), inhibiting the maturation and survival of autoantibody-producing B cells, which play a driving role in SLE pathogenesis. The randomized phase III BLISS-LN trial demonstrated that belimumab added to standard therapy improved renal response to treatment and reduced the risk of kidney-related events (HR 0.51), compared to standard therapy alone [52]. A post hoc analysis of this trial confirmed the risk reduction of LN flare and eGFR decline with belimumab [53]. The use of belimumab with glucocorticoids plus mycophenolate or low-dose cyclophosphamide has been recommended as a possible first-line treatment for Class III/IV LN [54].

### 4.2. Complement Inhibitors

The complement system is an integral part of innate immunity and plays a key role in host defense and tissue clearance of damaged cells and immunocomplexes. Three main activation pathways have been described: “classic”, “lectin” (LP), and “alternative” (AP), which converge in the cleavage of C3 factor into its active form by a C3-convertase and proceed in a common “terminal pathway” (TP). Pathological stimulation of these pathways has been linked to a broad spectrum of kidney diseases [55]. Over recent decades, the evolving comprehension of complement system involvement in the pathogenesis of kidney diseases has led to the development of targeted therapies designed to modulate aberrant activation, acting as “immunomodulators” rather than “immunosuppressors” [56]. The following is a brief description of anti-complement therapy currently approved or in late-stage clinical development (also summarized in Table 1).

#### 4.2.1. Targeting LP (Low-Grade Evidence)

Binding of mannose-binding lectin (MBL), ficolins, and collectin-11 to glycosylated antigens—both on pathogens and abnormal Ig—initiates this pathway through the recruiting of specific serine-proteases (MASP1-2) which cleave C4 and C2 factors, resulting in the formation of C4bC2a complex, namely the C3-convertase of LP [55].

Numerous data have highlighted the role of LP in the pathogenesis of IgA Nephropathy (IgAN). Up to 50% of kidney biopsies show mesangial staining of MBL together with IgA1; furthermore, elements such as MASP1-2 and C4d are usually co-localized. Glomerular deposition of these factors has been associated with more severe histological damage and worse kidney outcome [57]. Only one inhibitor has been tested so far: Narsoplimab, a fully human monoclonal antibody against MASP-2, the effector enzyme of LP. Although the Phase-II trial on 16 high-risk IgAN patients [58] initially described an anti-proteinuric effect, the Phase-III RCT [ARTEMIS-IgAN] did not meet its primary endpoint in proteinuria reduction.

#### 4.2.2. Targeting AP (Low-Grade Evidence)

AP is constitutively active by spontaneous low-level hydrolysis of C3 (“tickover mechanism”), which allows its interaction with FactorB and FactorD to form a temporary C3-convertase complex that promotes the generation of C3b, the effector of AP. The presence of microbial products triggers further amplification via the cleavage of loop C3. An adequate level of AP activation relies on a balance between positive (e.g., Properdin) and negative (e.g., Factor H and Factor I) regulator proteins. Dysregulation of this delicate system—both congenital and acquired—is a key pathogenetic factor in C3-glomerulpathy (C3G), immune complex-mediated membranoproliferative glomerulonephritis (IC-MPGN), and atypical hemolytic–uremic syndrome (aHUS) and results in tissue damage in many other kidney diseases, such as IgAN and ANCA-associated vasculitis (AAV) [55].

Drugs targeting C3 convertase formation or function have reached late-stage clinical trials:-Danicopan is a small molecule acting as a Factor D inhibitor tested in two phase-II trials for C3G and IC-MPGN, respectively, showing incomplete and unsustained AP inhibition due to pharmacokinetic/pharmacodynamic limitations [59]. A similar agent, Vemircopan, is currently being studied in a phase-II trial enrolling patients with IgAN and lupus nephritis (LN).-Iptacopan is designed to target factor B, inhibiting the formation of both C3 and C5 convertases, thus suppressing AP and TP. Promising antiproteinuric effect from phase-II studies in IgAN [60] and C3G [61] led to four phase-III trials starting, respectively, for IC-MPGN [APPARED], C3G [APPEAR-C3G], IgAN [APPLAUSE-IgAN], and aHUS [APPELHUS].-Pegcetacoplan, a direct inhibitor of C3 and C3b, was tested in an open-label phase-II trial [DISCOVERY] on 21 patients affected by complement-mediated kidney diseases (including eight C3Gs), showing a favorable 50% proteinuria reduction at 48 weeks in the C3G subgroup [62]. Results of another phase-II trial [NOBLE] on post-transplant recurrent C3G/IC-MPGN are expected soon, and a phase-III RCT [VALIANT] is ongoing with approximately 90 patients with both native and recurrent disease.

#### 4.2.3. Targeting TP

As stated previously, the constitution of a complex with C3-convertase activity is the core of complement activation regardless of the pathway originally involved. Further incorporation of C3b into existing C3-covertases leads to the formation of a C5-convertase, which marks the initiation of TP. Cleavage of C5 ultimately promotes the assembling of effector cytolytic Membrane Attack Complex (MAC). The concomitant inflammatory response occurs when anaphylatoxin C3a and C5a, generated through cleavage of C3 and C5, bind to specific receptors (e.g., C5aR) on leukocytes [55].

The first complement inhibitor tested and then approved in nephrology was DB01257, a monoclonal antibody targeting C5, commonly known as Eculizumab. Its use marked a milestone in the management of previously prognostic-unfavorable complement-mediated diseases such as aHUS.

Drugs with high-grade evidence:-Following the legacy of Eculizumab and preserving the target epitope, Ravulizumab was engineered to increase antibody half-life (52 vs. 11 days) through a “re-cycling” mechanism. In this way, the dosing interval was extended to 8 weeks, enhancing treatment adherence. Preliminary studies on aHUS demonstrated efficacy and safety in adults naïve to complement therapy [63], and in children either naïve [64] or previously treated with eculizumab [65]. Patients who switched to ravulizumab maintained stable kidney function and blood count. Since a direct head-to-head comparison of both anti-C5 antibodies would have been limited due to the rarity of the disease, a retrospective propensity-matched analysis was performed; outcome comparison between naïve adult patients treated with ravulizumab and a matched cohort from eculizumab trials did not show a significant difference [66]. Beyond approved use in aHUS, ravulizumab is being tested for both IgAN and LN in a phase-II study [SANCTUARY].-Avacopan is an antagonist of the neutrophil C5a receptor, the activation of which is involved in the pathogenesis of AAV. A randomized phase III trial [ADVOCATE] in 331 patients compared avacopan to tapering doses of prednisone in addition to standard-of-care induction therapy (Rituximab or Cyclophosphamide) for new-onset or relapsing AAV, with disease remission at week 26 being the primary outcome. Non-inferiority in remission at 26 weeks (73% vs. 70%) and superiority in sustained remission at 52 weeks (66% vs. 55%) were achieved, leading to regulatory approval [67]. Furthermore, the 2024 KDIGO Guidelines for Management of AAV [68] suggest avacopan to be an alternative to glucocorticoids for induction of remission.

Avacopan also underwent successful phase II testing in 71 patients with C3G [ACCOLADE trial], demonstrating reduction in progression of chronic lesions at protocol biopsy compared to placebo [69].

Drugs with low-grade evidence:-Crovalimab is another anti-C5 antibody, designed to allow subcutaneous rather than intravenous administration. Phase-III trials on aHUS [COMMUTE-A, COMMUTE-P] are currently recruiting.-Cemdisiran is an RNA interference therapeutic (see Section 6) designed to inhibit hepatic C5 synthesis. Despite the positive results from a phase II trial on 31 IgAN patients [70], no further developments have been planned.

### 4.3. Targeting B-Cell Dysregulation: APRIL System Inhibitors (Low-Grade Evidence)

A proliferation-inducing ligand (APRIL) is a growth factor of the tumor necrosis factor (TNF) super family involved in the proliferation and differentiation of the B cells. Recent evidence indicates a role of for APRIL in the pathogenesis of IgAN, thought to be responsible for the B cell dysregulation, IgA isotype switching, and overproduction of nephrotoxic galactose-deficient IgA1 (Gd-IgA1). Elevated serum levels of APRIL are found in patients with IgAN when compared to controls and have been associated with higher plasmatic Gd-IgA1, hence poor prognosis. The previously described BAFF is a member of the same superfamily as APRIL. Despite both factors exhibiting a partial structural analogy and an overlapping receptor affinity, a slightly different biological role in the regulation of B cells is hypothesized [71].

Anti-APRIL therapies are currently being investigated in clinical trials. Early findings indicate decreased Gd-IgA1 levels and proteinuria, suggesting for the first time a potentially disease-modifying role of these agents in IgAN.

-Atacicept and Telitacicept are dual BAFF/APRIL inhibitors and have reported a possible anti-proteinuric effect in randomized placebo-controlled phase 2 trials, JANUS [72] and NCT04291781, respectively [73]. The primary analysis of the phase 2b ORIGIN trial of atacicept on 116 IgAN showed a significant mean uPCR reduction from baseline in the pooled atacicept arms compared to placebo (31% vs. 7%, delta 25%, *p* = 0.037) [74].-Sibeprenlimab is a monoclonal antibody that neutralizes APRIL. In a double-blind, placebo-controlled, parallel-group phase 2 trial, 155 patients with IgAN were randomized in a 1:1:1:1 ratio to receive sibeprenlimab at increasing doses (2–4–8 mg/Kg) on top of supportive therapy or placebo. In terms of reduction in proteinuria from baseline, a significant linear dose effect was observed at 12 months (24 h-based-uPCR 47.2 ± 8.2%, 58.8 ± 6.1%, 62.0 ± 5.7%, 20 ± 12.6%), reaching the primary endpoint. The annualized eGFR slope seemed to be attenuated compared to placebo (−4.1 ± 1.7, 0.1 ± 1.6, −0.8 ± 1.6, −5.9 ± 1.7) [75]. A phase 3 RCT is underway [VISIONARY].

### 4.4. B-Cell Depletion Therapy (Anti-CD20)

The approval of Rituximab—a monoclonal antibody targeting CD20—in 1997 for non-Hodgkin lymphoma marked a milestone in the treatment of B-cell-related disorders, such as hematological malignancies and autoimmune diseases, due to its ability to induce sustained depletion of CD-20 cells. The use of Rituximab is currently recommended by 2021 KDIGO Glomerular diseases guideline [76] as a feasible first line therapy in primary MN, AAV, and class III/IV lupus nephritis, and can be considered when approaching steroid-resistant nephrotic syndrome (MCD or FSGS), IgA-vasculitis, anti-glomerular basement membrane disease, and other rapid progressive glomerulonephritis.

Resistance to Rituximab and the development of anti-drug antibodies led to novel anti-CD20 drugs that have recently gained approval: ofatumumab and ublituximab for relapsing multiple sclerosis and obinutuzumab for chronic lymphocytic leukemia and follicular lymphoma. The effector mechanisms on targeted CD-20 cells vary depending on the generation, structural aspects, and binding sites of the antibody (Table 2) [77]. Their use in current nephrological practice is off-label and limited to little low-grade evidence, such as the following:-A case series by Podestà et al. [78] described the use of Ofatumumab as rescue treatment in ten rituximab-resistant and seven rituximab-intolerant patients affected by primary MN, showing greater remission of proteinuria in intolerant patients. Similarly, Haarhaus et al. [79] reported the successful use of Ofatumumab in four refractory LN patients who developed an infusion reaction to rituximab at re-treatment.-The effect of Obinutuzumab on proliferative LN was assessed in a phase II trial [NOBILITY] on 125 patients randomized to receive Obinutuzumab or placebo in combination with standard therapies. Complete renal remission at 104 weeks was greater in the treatment group compared to placebo (41% vs. 23%, *p* = 0.026) despite the fact that the primary endpoint at 52 weeks was not statistically significant [80]. In a post hoc analysis, Obinutuzumab demonstrated superiority in preservation of kidney function and prevention on LN flares compared to SoC [81]. A global phase-III study is ongoing [NCT04221477].

### 4.5. Daratumumab (Low-Grade Evidence)

Abnormal immunoglobulin production by plasma cell clones is involved in the pathogenesis of a wide range of kidney diseases, and the role of the anti-CD38 antibody daratumumab has been investigated. The use of daratumumab in refractory lupus nephritis (LN) has been reported to be associated with significant clinical and serologic responses (reduction in urine protein, serum creatinine levels, and a reduction in anti-double-stranded DNA of about 50–60%) [82,83]. Similarly, experiences of daratumumab in refractory antineutrophil cytoplasmic antibody (ANCA)-associated vasculitis (AAV) were limited, but clinical reports have shown an effective remission of AAV with stabilization of kidney function [84]. Furthermore, the use of daratumumab in light chain amyloidosis (AL amyloidosis) has provided encouraging results; the phase III ANDROMEDA study [85] revealed that the addition of daratumumab to the standard of care was associated with better hematologic, renal, and cardiac responses. Further phase III clinical studies of daratumumab in LN, AL amyloidosis with renal involvement, and AAV are warranted.

### 4.6. Endothelin Receptor Antagonists (High-Grade Evidence)

It has been demonstrated that endothelin plays a central role in the regulation of glomerular filtration and in acid–base and hydro-sodium balances; endothelin increases in conditions of acidosis and hyperglycemia and in the presence of inflammatory cytokines and insulin, and causes sustained vasoconstriction of afferent arterioles, leading to hyperfiltration and proteinuria [86]. Therefore, the use of endothelin receptor antagonist (ERA) has been studied as a strategy to reduce protein loss with urine and, eventually, to slow the progression of kidney disease. Interim results from the ongoing phase 2 AFFINITY study [87] revealed the safety and efficacy of the ERA atrasentan in reducing proteinuria in patients with proteinuric glomerular diseases. Similarly, interim analysis of the phase 3 ALIGN study [88] demonstrated superiority of atrasentan versus placebo in reducing proteinuria in patients with IgA nephropathy receiving standard of care (RAS inhibitor). In 2023, the first-in-class dual endothelin–angiotensin receptor antagonist sparsentan received approval in the US for IgAN, based on the milestone results from the phase 3 PROTECT trial, which demonstrated the clinically meaningful reduction in proteinuria and stability of kidney function, compared to irbesartan [89]. Moreover, sparsentan was shown to be superior to irbesartan in reducing proteinuria, even in patients with focal segmental glomerulosclerosis (DUPLEX trial), however, without significant differences in the slope of eGFR [90].

## 5. Kidney Transplantation

Renal replacement therapies have seen a change in epidemiology in recent years, with shorter duration of dialysis time and improved quality of life for patients undergoing dialysis. In the same period, there has been an increase in the rate of kidney transplantation, a type of renal replacement therapy that offers the best prognosis and quality of life when compared with peritoneal and hemodialysis [91]. This trend stimulated research around the treatment of the main complications of KT, and in particular, acute and chronic rejection, which are considered the most fearsome and frequent medical complications after KT. The result is a continuous innovation in terms of comprehension of mechanisms of damage and novel drugs to treat the aforementioned complications.

### 5.1. Imlifidase (Low-Grade Evidence)

One of the most intriguing challenges in kidney transplantation is overcoming the obstacle of hyperimmunization, which consists of the presence of high levels of donor-specific antibodies (DSAs) against human leukocyte antigens (HLAs) in the candidate recipient serum, which limits the possibility of transplantation due to the extremely high risk of early humoral rejection. In 2020, European Medicines Agency has approved imlifidase as a desensitization agent in highly sensitized patients (≥80% panel reactive antibody) with a positive cross-match to a deceased kidney donor. Imlifidase is a recombinant protease derived from Streptococcus pyogenes that cleaves all four subclasses of human IgG into F(ab′)2 and Fc fragments, inhibiting Fc-mediated complement-dependent cytotoxicity (CDC) and antibody-dependent cellular cytotoxicity (ADCC) [92]. Within a few hours after administration, elimination of DSA allows for obtaining a negative cross-match and therefore enables transplantation with HLA-incompatible donors. Recipients of imlifidase-enabled allografts showed comparable outcomes to that of other highly sensitized patients who underwent HLA-incompatible transplantation at 3 years post-transplantation (84% graft survival and 38% antibody-mediated rejection rate) [93].

### 5.2. Daratumumab (Low-Grade Evidence)

Daratumumab is a human IgG1κ monoclonal antibody directed against CD38, a transmembrane glycoprotein highly expressed on the surface of immune cells such as plasmacells and natural killer cells. Targeting CD38 causes depletion of CD38-expressing cells via apoptosis, complement-dependent cytotoxicity, and antibody-dependent cellular cytotoxicity. Although the use of daratumumab as treatment for multiple myeloma is widely consolidated, its use in kidney transplantation to reduce nonmalignant HLA antibody-producing plasmacells needs further research [94]. The presence of donor-specific antibodies is critical in antibody-mediated rejection, and depletion of alloantibody-producing plasmacells may be an effective therapeutic strategy for recipient desensitization prior to kidney transplantation [95] and for chronic active antibody-mediated rejection [96].

### 5.3. Tocilizumab (Low-Grade Evidence)

Kidney transplant graft survival depends on several factors, among which immunological complications contribute significantly to both short- and long-term damage. In the pathway of antibody-mediated rejection (AMR), the cytokine interleukin 6 (IL-6) plays a key role in the regulation of the inflammatory process and in the stimulation of T and B cells and plasma cells. Initially approved for the treatment of rheumatoid arthritis and juvenile idiopathic arthritis, the IL-6 receptor antagonist Tocilizumab was subsequently proposed as a therapeutic strategy to treat AMR in kidney transplantation and to desensitize highly HLA-sensitized kidney transplant candidates. In fact, inhibition of IL-6 receptor leads to a decrease in the production of alloantibodies by activated B cells [97] and a reduction in inflammation through the regulation of T cell response [98]. Although clinical experiences in small studies have demonstrated a reduction in donor-specific antibodies and interstitial inflammation and a stabilization of long-term graft function in patients with chronic active AMR [99], results from large phase 3 clinical trials are lacking [NCT04561986].

### 5.4. Belatacept (High-Grade Evidence)

Calcineurin inhibitor (CNI) nephrotoxicity is recognized to affect the long-term outcomes of kidney transplants; chronic endothelial injury and arteriolar vasoconstriction lead to an irreversible and progressive decline in allograft function [100]. In order to increase allograft survival, a new non-nephrotoxic immunosuppressive regimen without calcineurin inhibitor was developed. Belatacept is a protein produced by the fusion of the Fc fragment of human IgG1 immunoglobulin and the extracellular domain of cytotoxic T-lymphocyte-associated antigen 4, and it can block the costimulation of T lymphocytes by the inhibition of the interaction between CD28 and CD80/86. Results from the studies BENEFIT and BENEFIT-EXT demonstrated that patient and graft survival with belatacept was comparable to cyclosporine, and long-term use of belatacept was associated with sustained improvement in renal function (higher GFR) versus cyclosporine [101]. However, a higher occurrence of acute rejection and post-transplant lymphoproliferative disorders was reported with belatacept; therefore, a low-intensive belatacept regimen was approved for use in EBV-seropositive patients [102].

## 6. Genetic Kidney Disease

Although it is widely accepted that genetics play a role in the pathogenesis of kidney diseases, their real contribution remains challenging to estimate, especially when dealing with multifactorial and/or polygenic nephropathies [103]. Monogenic kidney diseases make up approximately 50% and 30% of non-diabetic CKD, respectively, in pediatric and adult cohorts, respectively [104], with more than 600 genes linked to single-gene disorders [105]. Recent findings in molecular genetics prompted new perspectives in assessing disease susceptibility and risk factors, as well as predicting diagnosis and therapeutic response in both monogenic and multifactorial afflictions [103].

### 6.1. RNA Interference (RNAi)

One of the most promising strategies relies on “RNAi interference”, an endogenous mechanism of post-transcriptional gene regulation described by 2006 Nobel Scientists Fire and Mello. RNAi-based therapeutics mimic this natural process by targeting a specific mRNA transcript through a recombinant complementary nucleotide sequence that inhibits gene expression, and hence protein synthesis. The interfering RNA can be administered intravenously (formulated inside a lipid nanoparticle) or subcutaneously (conjugated with Gal-NAC residues), and is engineered to be taken up by hepatocytes that arrest the production of a specific liver-derived protein [106].

RNA-therapy has gained increasing attention over the last decade as a potential weapon against genetic diseases, as stated by the high number of drugs being tested in clinical trials. Moreover, the large-scale production of a vaccine against severe acute respiratory syndrome coronavirus-2 (SARS-CoV-2) during the recent pandemic brought this technology into the spotlight [107].

#### 6.1.1. Rare Nephrolithiasis (High-Grade Evidence)

Primary Hyperoxaluria (PH) is a rare autosomal-recessive disorder characterized by abnormal liver production of oxalate, leading to hyperoxaluria, nephrolithiasis, and nephrocalcinosis, with subsequential progressive kidney failure and systemic oxalosis. Three forms are classified based on the affected enzyme: AGTX in PH1, GRHPR in PH2, and HOGA1 in PH3. PH1 is the most common type, with heterogeneous clinical presentation (ranging from neonatal kidney failure to adult nephrolithiasis) and poor kidney outcome despite supportive therapy [108]. Until recent years, the only available etiological treatment in pyridoxine-unresponsive forms was liver transplantation, eventually combined with kidney transplantation (either sequential or concomitant) in case of renal failure [109]. Targeting specific enzymes involved in oxalate metabolism using RNAi is an innovative strategy in PH. This approach has shown to lower endogenous oxalate production, by-passing dysfunctional metabolic pathways:-Lumasiran, which targets glycolate oxidase production, was the first specific treatment approved for patients with PH1 after the results of ILLUMINATE phase III trials, which demonstrated significant reduction in urine and plasma oxalate irrespective of age and residual kidney function [110].-Similarly, Nedosiran, another recently approved drug designed to inhibit lactate-dehydrogenase production, reduced urine oxalate level in PH1 patients older than 9 years with preserved kidney function (eGFR > 30 mL/min) [111].

Although the long-term safety and efficacy of these drugs need to be investigated, early results are encouraging and indicate a potential “paradigm shift” in the treatment of PH1, as recently stated in a consensus statement from ERK-Net and Oxal-Europe [112].

#### 6.1.2. Beyond Kidney Stones 

Many other RNAi-based therapeutics are surfacing in the current landscape. Among those supported by high-grade evidence, we report the following:-Patisiran, an RNAi agent that inhibits hepatic production of transthyretin, was approved in 2018 for the treatment of hereditary transthyretin-mediated amyloidosis (hATTR), since the APOLLO phase-III trial [113] showed improvement of disease-related manifestations such as neuropathy and exercise intolerance. Vutisiran was later designed for subcutaneous administration and tested in phase-III HELIOS-A trial [114], gaining regulatory approval.-Inclisiran is a lipid-lowering RNAi drug approved in 2021 for clinical atherosclerotic cardiovascular disease (ASCVD) and heterozygous familiar hypercholesterolemia (HeFH). Inclisiran suppress PCSK9 production, a circulating protein that promotes degradation of low-density lipoprotein receptor (LDL-R), resulting in LDL clearance [115].

Noteworthy other drugs gained low-grade evidence in trials:
-Teprasiran is engineered to inhibit p53-mediated cell death, a key pathogenetic process in ischemia reperfusion-induced AKI. A randomized, placebo-controlled, double-blind phase-II study on 360 high-risk patients undergoing cardiac surgery showed a reduction in post-operative AKI incidence, severity, and duration after drug administration [116]. Teprasiran is also under investigation for the prevention of delayed graft function in kidney transplant recipients from diseased donors [NCT0080234, NCT02610296].-Zilebesiran, an investigational RNAi therapeutic that inhibits hepatic angiotensinogen synthesis, has been recently tested as a blood pressure-lowering agent in a phase-I study [117].

## 7. Conclusions

We have provided a synthetic overview of the treatments available for kidney diseases at the beginning of 2024 (Figure 1) and which we refer to the volume for a more in-depth discussion of the specific drug classes. From the review of the literature, it is evident that steps forward have been made in most of the branches of nephrology. Mineralocorticoid receptor antagonists, SGLT2 inhibitors, and oral HIF stabilizers will help to delay CKD progression. Drugs acting on the terminal activation of complements proved their efficacy in the first studies enrolling patients with IgAN, lupus nephritis, and ANCA-associated vasculitis, these being immune-mediated diseases of significant prevalence in the nephrology patient population. Endothelin blockade also helped to manage patients with IgAN. In kidney transplant patients, several drugs may be implemented to improve desensitization before transplant, whereas robust evidence is already available for the use of belatacept as an immunosuppressor alternative to the classic CNI in post-transplant follow-up. Genetic therapies also made progress in kidney diseases. RNA interference treatments have shown efficacy in reducing plasma and urine oxalate levels in primary hyperoxaluria and improved the management of hereditary transthyretin-mediated amyloidosis.

What, thus, emerges from the current literature is that nephrologists now have more tools to manage and improve the prognosis of patients with kidney diseases. Moreover, compared with the past, we may also note an improved quality of evidence for each drug, with intervention and randomized trials well conducted. Let us leave it to the future and daily clinical work to monitor and evaluate what has been accomplished so far. The picture of CKD is changing, and this disease is now no longer considered an inexorably progressive condition.

## Figures and Tables

**Figure 1 biomedicines-12-00828-f001:**
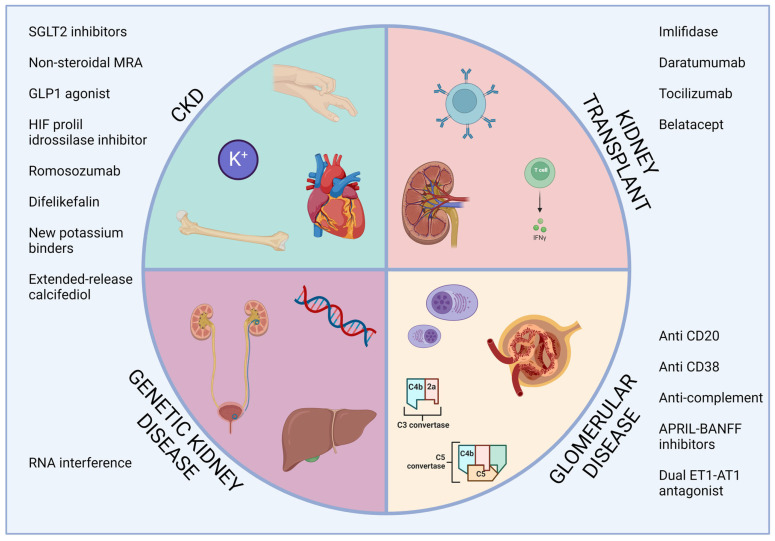
Overview of the available treatments of kidney disease. APRIL, a proliferation inducing ligand; CKD, chronic kidney disease; ET1-AT1, endothelin 1—angiotensin II receptor type 1; GLP1, glucagon-like peptide 2; HIF, hypoxia-inducible factor; MRA, mineralocorticoid receptor antagonist; RNA, ribonucleic acid; SGLT2, sodium-glucose cotransporter 2.

**Table 1 biomedicines-12-00828-t001:** Brief overview of complement inhibitor currently approved and in late-stage clinical development for kidney diseases.

Targeted Pathway	Drug	Type of Inhibitor	Inhibition Target	Clinical Trial Phase	Target Kidney Diseases
Lectin	Narsoplimab	mAb	MASP-2	III (failed)	IgAN
Alternative	Danicopan	Small molecule	Factor D	II	C3G, IC-MPGN
	Vemircopan	Small molecule	Factor D	II	IgAN, LN
	Iptacopan	Small molecule	Factor B	III	IgAN, C3G, IC-MPGN, aHUS
	Pegcetacoplan	Pegylated pepetide	C3	III	C3G, IC-MPGN
Terminal	Eculizumab	mAb	C5	Approved	aHUS
	Ravulizumab	mAb	C5	II Approved	IgAN, LN aHUS
	Crovalimab	mAb	C5	III	aHUS
	Cemdisiran	siRNA	C5	II (no further development)	IgAN
	Avacopan	Small molecule	C5aR	II–III Approved	C3G AAV

MASP-2, mannose-binding protein-associated serine protease 2; mAb, monoclonal antibody; IgAN, IgA-Nephropathy; C3G, C3-glomerulopathy; IC-MPGN, immune-complex-mediated membranoproliferative glomerulonephritis; aHUS, atypical hemolytic uremic syndrome; AAV, antineutrophil cytoplasmic antibody-associated vasculitis; LN, lupus nephritis; siRNA, silencing-RNA; C5aR, C5a receptor.

**Table 2 biomedicines-12-00828-t002:** Brief overview of anti-CD20 antibodies and their main effector mechanisms.

	Rituximab	Ofatumumab	Ublituximab	Obinutuzumab
Generation	I	I	I	II
Type	Chimeric	Humanized	Chimeric	Humanized
ADCC	Intermediate	Intermediate	Very high	Very high
Direct Cytotoxicity	Low	Low	Intermediate	Very high
CDC	Intermediate	High	Intermediate	Low
ACDP	Intermediate	High	Very High	Very High

ADCC, antibody-dependent cellular cytotoxicity; CDC, complement-dependent cytotoxicity; ACDP, antibody-dependent cellular phagocytosis.

## Data Availability

No new data were created or analyzed in this study. Data sharing is not applicable to this article.

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
