# Peer review of "Improving Kidney Disease Care: One Giant Leap for Nephrology"

_biomedicines, 2024, doi:10.3390/biomedicines12040828_

Round 1

Reviewer 1 Report

Comments and Suggestions for Authors

The presented manuscript provides a comprehensive overview of the emerging therapeutic modalities for various kidney associated diseases. While the idea is interesting and useful for the wide range of clinicians, there little is said about the personalised approach, since the manuscript describes mostly the pharmacological treatment options and says little about how to provide this option to the right patient in the right moment, which are the essential questions for personalised approached. Moreover, little is said about biomarkers which could also help regarding the treatment decisions. Therefore, this should be added to the manuscript, or otherwise the title should be changed. Moreover, little is said about the methodology of the review. More specifically, this looks like the narrative review, but nothing is said about the inclusion or exclusion criteria and other important aspects for this type of study. Finally, there is little critical appraisal for various modalities and studies. 

Comments on the Quality of English Language

The quality of language is poor, both semantically and grammatically. 

Here are several examples:

Line 25: Herein, we here provide an overview of the principal drug classes with proved efficacy in the context of the aforementioned clinical conditions. After "herein" there is no need for "here".

Line 35 The aforementioned importance of CKD is due to the fact that, its onset confers to each affected individual a

strikingly high risk of cardiovascular (CV) events, hospitalizations, mortality for all-causes and other major outcomes, including kidney failure (KF) namely the final stage of kidney disease which requires the need for planning renal replacing therapies such as kidney transplantation or dialysis (2). The sentence is too long, hard to follow and incorrect (eg "including" and afterwards "namely". 

Line 58 The aim of the present review is to summarize the recent discoveries in terms of drugs that target a specific kidney disease or a specific pattern of damage, conferring an individual protection against CV and renal risk and how these new findings are changing nephrology from an uncertain matter to a systematic and more organized discipline. How was nephrology "uncertain matter"?

Reviewer 2 Report

Comments and Suggestions for Authors

I have read with interest the manuscript "Personalizing care in Nephrology: on the road". I support undertaking such a task, as a concise overview of the principal drug classes emerging in treating kidney diseases, is both needed and interesting. The text is clear, well organized and accompanied by informative tables.

I have a few comments that authors could kindly consider as helpful.

1. The title and abstract are misleading. The manuscript provides no clues on how to personalize care using the new therapeutic options. The aim of the study is a summary of new therapies. It is not a guide for clinical nephrologists, despite the claims in the abstract.

2. I suggest dividing existing sections into therapy in use (supported by RCTs) and experimental therapies with low-grade evidence. This will add a lot of clarity. 

3. The same as above applies to the Figure. For me, it is confusing, to suggest a similar role as a pillar for SGLT2 inhibitors and Romosozumab. The whole picture does not make much sense and should be reconsidered.

4. I wonder why the authors choose not to include patiromer in a CKD section and tolvaptan in a genetic kidney disease one.

5. Finally, the conclusions are too vague, and general, not highlighting the main points of the manuscript.

Reviewer 3 Report

Comments and Suggestions for Authors

The authors present a feature paper about personalizing care in nephrology and the use of novel drugs to treat kidney disease and its complications. The work generally is interesting summarizing the clinical data as to the medical management of kidney disease. Some comments follow: 

1) The manuscript would benefit with a small section in the introduction regarding the main pharmacological targets in kidney disease. What disturbances due to reduced kidney function require pharmacological approaches. 

2) Another point that could be further elaborated is wheather CKD or relative disorders the cause or the result of co-existing conditions. How kidney diseases affect organ function and lead in further pathophysiological conditions. 

3) The document mentions several clinical trials. Considering the personalized medicine approaches and the available evidence, are there any updates in th regulatory guideline as to the management of kidney diseases? There are some reports from the FDA as to approvals but maybe some further discussion could be added. 

4) There are some references to be discussed considering that these patient represent a special population group that require additional approaches towards optimization of therapeutic approaches

I) Int J Environ Res Public Health. 2020 Dec 6;17(23):9101. doi: 10.3390/ijerph17239101

II) Int J Mol Sci. 2017 Jun 10;18(6):1248. doi: 10.3390/ijms18061248.

III) Clin J Am Soc Nephrol. 2018 Jul 6;13(7):1085-1095. doi: 10.2215/CJN.00340118.

Comments on the Quality of English Language

n/a

Reviewer 4 Report

Comments and Suggestions for Authors

I considered the manuscript entitled “Personalizing care in Nephrology: on the road” by

Michele Provenzano, et al, which is intended to be published in Biomedicines journal.

As authors say at the conclusions: “We provided a synthetic overview of the available treatments of kidney diseases at the beginning of 2024”. This is the real value of the present manuscript. Authors take advantage of a rapid writing, gathering the new drugs available at this moment to treat renal pathologies. This appears interesting and of informative interest for both, young nephrologist in training and for old nephrologists who need updating. However the biggest drawback is the writing, their sentence should appear more elegant: “We provide a summary overview of treatments available for kidney diseases in early 2024”.

Line 25, “Herein, we here provide an overview of the principal drug classes with proved

efficacy in the context of the aforementioned clinical conditions.”, rewrite it is not understandable.

Line 36, “is due to the fact that, its onset confers to each affected” please rewrite.

Line 38, “including kidney failure (KF) namely the final stage of kidney disease which requires the need for planning renal replacing therapies such as

kidney transplantation or dialysis” too long the paragraph

Line 41, ” CKD that are almost doubles in the past” rewrite

Line 43, “This also makes us reflect on the evidence that common diseases such as hypertension, diabetes and obesity may damage the kidney, thus worsening in an extremely negative manner the quality of life and the individual prognosis” not understandable, unnecessary, too long.

Line 46, “As already stated, CKD is a chronic condition” It is obvious that chronic kidney disease is a chronic condition…

…….

I don't keep correcting the writing of the manuscript... It needs a specialized English medical writer to introduce a more professional appearance of the final version.

Comments on the Quality of English Language

The english writing is not fine, it needs a professional correction. Apart, the manuscript needs a Senior writer with experience in the field to change the format of the writing and give the wording a better look

Round 2

Reviewer 1 Report

Comments and Suggestions for Authors

The authors have successfully responded to all of the reviewers comments. 

Comments on the Quality of English Language

Some minor edits are still needed. 

Reviewer 4 Report

Comments and Suggestions for Authors

none